# Impact of Administering Intravenous Azithromycin within 7 Days of Hospitalization for Influenza Virus Pneumonia: A Propensity Score Analysis Using a Nationwide Administrative Database

**DOI:** 10.3390/v15051142

**Published:** 2023-05-10

**Authors:** Takatomo Tokito, Takashi Kido, Keiji Muramatsu, Kei Tokutsu, Daisuke Okuno, Hirokazu Yura, Shinnosuke Takemoto, Hiroshi Ishimoto, Takahiro Takazono, Noriho Sakamoto, Yasushi Obase, Yuji Ishimatsu, Yoshihisa Fujino, Kazuhiro Yatera, Kiyohide Fushimi, Shinya Matsuda, Hiroshi Mukae

**Affiliations:** 1Department of Respiratory Medicine, Nagasaki University Graduate School of Biomedical Sciences, Nagasaki 852-8501, Japan; 2Department of Preventive Medicine and Community Health, University of Occupational and Environmental Health, Japan, Kitakyushu 807-0804, Japan; 3Department of Infectious Diseases, Nagasaki University Graduate School of Biomedical Sciences, Nagasaki 852-8501, Japan; 4Department of Nursing, Nagasaki University Graduate School of Biomedical Sciences, Nagasaki 852-8501, Japan; 5Department of Environmental Epidemiology, Institute of Industrial Ecological Science, University of Occupational and Environmental Health, Japan, Kitakyushu 807-0804, Japan; 6Department of Respiratory Medicine, University of Occupational and Environmental Health, Japan, Kitakyushu 807-0804, Japan; 7Department of Health Policy and Informatics, Graduate School of Medical and Dental Sciences, Tokyo Medical and Dental University, Japan, Tokyo 113-8519, Japan

**Keywords:** azithromycin, seasonal influenza virus, COVID-19, coronavirus disease, acute respiratory distress syndrome, infection, inflammation, diagnosis procedure combination, inverse probability of treatment weighting

## Abstract

The potential antimicrobial and anti-inflammatory effectiveness of azithromycin against severe influenza is yet unclear. We retrospectively investigated the effect of intravenous azithromycin administration within 7 days of hospitalization in patients with influenza virus pneumonia and respiratory failure. Using Japan’s national administrative database, we enrolled and classified 5066 patients with influenza virus pneumonia into severe, moderate, and mild groups based on their respiratory status within 7 days of hospitalization. The primary endpoints were total, 30-day, and 90-day mortality rates. The secondary endpoints were the duration of intensive-care unit management, invasive mechanical ventilation, and hospital stay. The inverse probability of the treatment weighting method with estimated propensity scores was used to minimize data collection bias. Use of intravenous azithromycin was proportional to the severity of respiratory failure (mild: 1.0%, moderate: 3.1%, severe: 14.8%). In the severe group, the 30-day mortality rate was significantly lower with azithromycin (26.49% vs. 36.65%, *p* = 0.038). In the moderate group, the mean duration of invasive mechanical ventilation after day 8 was shorter with azithromycin; there were no significant differences in other endpoints between the severe and moderate groups. These results suggest that intravenous azithromycin has favorable effects in patients with influenza virus pneumonia using mechanical ventilation or oxygen.

## 1. Introduction

Seasonal influenza viruses cause a seasonal epidemic, with around 290,000 to 640,000 respiratory deaths annually [1,2,3]. In contrast to those during the coronavirus disease (COVID-19) pandemic period, the number of patients with influenza has recently decreased. However, the number of patients with influenza increased in some areas, and we should be concerned about pandemic influenza and the possibility of a “twindemic” with COVID-19 [4,5,6]. Severe cases of influenza, similar to COVID-19, can cause excessive inflammation, resulting in acute respiratory distress syndrome (ARDS) and death [7]. The clinical efficacy of immunosuppressive therapy such as steroids, Janus kinase inhibitors, and anti-interleukin (IL)-6 antibody therapy has been shown in COVID-19 patients with respiratory failure, renewing interest in their efficacy in viral infection [8,9,10]. However, although antiviral agents have been established for influenza, treatment for excessive inflammation and severe respiratory failure has not been established [11,12]. Moreover, the effects of corticosteroids have been reported to worsen prognosis in influenza [13,14]. Thus, establishing other beneficial immunomodulatory treatments for patients with severe influenza is necessary.

Azithromycin, a macrolide antibiotic, has favorable effects on critical respiratory diseases owing to its antimicrobial and anti-inflammatory effects. A meta-analysis showed that beta-lactam plus macrolide outperformed beta-lactam alone, with an odds ratio (OR) of 0.80 for all-cause death, for community-acquired pneumonia (CAP) [15]. A secondary analysis of a multicenter randomized controlled trial (RCT) of patients with ARDS showed that macrolide use was associated with lower 180-day mortality (hazard ratio (HR), 0.46) [16]. Furthermore, an RCT including patients with influenza showed that inflammatory cytokine levels decreased more rapidly in the oseltamivir/azithromycin group than in the oseltamivir monotherapy group [17]. Therefore, azithromycin may have favorable effects in patients with severe influenza virus pneumonia, but its effectiveness has not yet been demonstrated.

As validation by a large prospective RCT is useful but time-consuming and costly, we retrospectively evaluated the effects of intravenous azithromycin administration in influenza virus pneumonia patients with respiratory failure, using the Diagnostic Procedure Combination (DPC) database, a nationwide Japanese administrative database. Although the DPC database is limited by being a retrospective study, it has advantages such as low financial and time costs, real-world data are available, and long-term drug use and evaluation can be easily performed. We used the inverse probability of treatment weighting (IPTW) method with estimated propensity scores to minimize the bias introduced by baseline characteristics in retrospectively collected data in the azithromycin and non-macrolide groups.

## 2. Materials and Methods

### 2.1. Study Design

The research methodologies used in this retrospective observational study were in accordance with the Strengthening the Reporting of Observational Studies in Epidemiology (STROBE) Statement [18].

### 2.2. Ethics 

The requirement for informed consent was waived owing to the retrospective nature of the study and removal of any personally identifiable information from the extracted data. This study was approved by the Ethics Committee of Medical Research, University of Occupational and Environmental Health, Japan (approval number: R2-007), and conducted according to the guidelines of the Declaration of Helsinki.

### 2.3. Data Source

The DPC is a Japanese case-mix patient classification system launched in 2002 for payment management and the modernization of the healthcare system [19]. It covered approximately 80% of all acute care inpatient hospitalizations in Japan in 2020 [13,20,21]. The database contains the following patient details: age, sex, diagnosis, comorbidities at admission and during hospitalization (coded according to the International Classification of Diseases, Tenth Revision (ICD-10)), state of consciousness according to the Japan Coma Scale, medical procedures, medications, intensive care unit (ICU) admission, interventional procedures (including hemodialysis, mechanical ventilation, and administration of heart–lung medication), length of hospital stay, and discharge status (including in-hospital death). Although the DPC database does not include influenza virus types, information on influenza virus types circulating seasonally in Japan is available in the Japanese National Institute of Infectious Diseases’ database (https://www.niid.go.jp/niid/en/influenza-e.html (accessed on 7 May 2023)). According to the data presented in this database, the respective rates of A/H1, A/H3, and B viruses were 4%, 78%, and 18% during 2016/2017 (September 2016 to August 2017) and 23%, 32%, and 45% during 2017/2018 (September 2017 to August 2018). Thus, A/H1, A/H3, and B viruses should be the major virus types present in our study from April 2016 to March 2018.

### 2.4. Patient Selection and Definitions

We selected patients with influenza virus pneumonia (ICD-10 code J10.0 or J11.0) or influenza (ICD-10 code J10.1 or J10.8) with ARDS (J80). Data of patients admitted between April 2016 and March 2018 who met the criteria were extracted from the DPC database. To simply observe the effects of administering intravenous azithromycin within 7 days of hospitalization on standard therapy, patients who received macrolide antibiotics other than intravenous and oral erythromycin, clarithromycin, josamycin, roxithromycin and spiramycin, and oral azithromycin within 7 days of hospitalization were excluded from this analysis. Patients who did not use oxygen therapy were classified into the oxygen-free group (mild), those who used oxygen therapy or nasal high-flow (NHF) oxygen therapy were categorized into the oxygen group (moderate), and those who used non-invasive positive pressure ventilation (NPPV) or invasive mechanical ventilation (IMV) were classified into the ventilator group (severe) within 7 days of hospitalization. The patients were also divided into the azithromycin and non-macrolide groups, with patients treated with intravenous azithromycin within 7 days of hospitalization being included in the azithromycin group.

### 2.5. Variables

The following variables were considered confounding factors: age (in years), sex, emergency admission, emergency transport, home healthcare, hospital volume (number of patients who met the inclusion criteria between April 2016 and March 2018), smoking, diagnosis of asthma, cancer, cardiovascular disease (diagnosis of acute myocardial infarction, cardiac valvular disease, cardiomyopathy, or pulmonary embolism), cerebrovascular disease (diagnosis of cerebral hemorrhage, subarachnoid hemorrhage, or cerebral infarction), chronic kidney disease, chronic respiratory failure, chronic obstructive pulmonary disease, diabetes mellitus, liver dysfunction (diagnosis of hepatitis, liver cirrhosis, or liver failure), neurological dysfunction (Japan Coma Scale score at admission ≥ 100 (indicating a response by closing eyes, no verbal response, and movement in response to pain) [22]), heart failure, hypertension, interstitial lung diseases, medication use (albumin, antithrombin III, heparin, immunoglobulin, insulin, transfusion of platelets or red blood cells, recombinant human soluble thrombomodulin (rhTM), sivelestat, vasopressors, oseltamivir, steroid, laninamivir, peramivir, carbapenem, cephalosporin (1st–4th generation), clindamycin, minocycline, new quinolone, penicillin antibiotics, or anti-methicillin-resistant *Staphylococcus aureus* (MRSA) drugs), emergency hemodialysis or maintenance hemodialysis, number of IMV wearing days, NPPV, NHF oxygen therapy, oxygen therapy, use of polymyxin B-immobilized fiber column, and ICU admission within 7 days of hospitalization. Intravenous administration of antibiotics was considered as a covariate; oral administration was not considered.

### 2.6. Outcomes

The primary endpoint was mortality (in-hospital mortality, 30-day mortality, and 90-day mortality). The secondary endpoints were the duration of ICU management, mechanical ventilation, and hospital stay throughout the hospitalization period.

### 2.7. Statistical Analyses

The propensity score was calculated using a logistic model with baseline variables that could affect azithromycin administration, including all the variables described above. The C-statistic (area under the operating characteristic curve) was used to evaluate the goodness of fit. Subsequently, we adjusted the covariates and compared the outcomes between the azithromycin and non-macrolide groups using the IPTW method [23]. Compared to other propensity score analyses, the IPTW method has the advantages of a larger effective sample size and lower impact from strong residual bias that can occur in sample selection by matching [24]. The covariates before adjustment were evaluated using the chi-squared (χ^2^) test for categorical variables and unpaired *t*-test for continuous variables. All analyses were conducted using STATA/IC 14.0 (StataCorp, College Station, TX, USA). Statistical significance was set at *p* < 0.05.

## 3. Results

Figure 1 depicts the flowchart of the patient selection. Data of 5359 patients with influenza virus pneumonia or influenza with ARDS were extracted from the DPC database as described in the Section 2. We excluded 293 patients treated with macrolides other than intravenous azithromycin, and 5066 patients were therefore included in this study. They were classified into three groups based on their respiratory status: 1720 mild patients (33.9%) without oxygen therapy, 2739 moderate patients (54.1%) with NHF oxygen therapy or oxygen therapy, and 607 severe patients (12.0%) with mechanical ventilation.

### 3.1. Mild Group

Of the 1720 mild patients, 17 (1.0%) were treated with intravenous azithromycin at 500 mg/day for >1 day, and 1703 (99.0%) were not treated with macrolides within 7 days of hospitalization. Cardiovascular disease, cerebrovascular disease, neurological dysfunction, maintenance hemodialysis, antithrombin III, rhTM, sivelestat, cephalosporin (1st, 2nd, and 4th generation), clindamycin, minocycline, anti-MRSA drugs, number of IMV weaning days, NPPV, NHF oxygen therapy, oxygen therapy, use of polymyxin B-immobilized fiber column, and ICU admission rate were excluded from the variables because there were no patients to include. Subsequently, we used the IPTW method with estimated propensity scores; the C-statistic of the propensity score was 0.823. However, we were unable to adjust for variables and outcomes between the azithromycin and non-macrolide groups using the IPTW method because of the small number of patients treated with intravenous azithromycin.

### 3.2. Moderate Group

Of the 2739 moderate patients, 85 (3.1%) were treated with intravenous azithromycin at 500 mg/day for >1 days, and 2654 (96.9%) were not treated with macrolides. The C-statistic for the propensity score was 0.804. Table 1 presents the patients’ baseline characteristics. Platelet transfusion, antithrombin III, sivelestat, immunoglobulin, polymyxin B-immobilized fiber column, NPPV, and IMV weaning days were excluded from the variables because there were no patients to include. The baseline variables including emergency admission (78.8% vs. 68.2%, *p* = 0.037), smoking (50.6% vs. 32.7%, *p* = 0.001), liver dysfunction (4.7% vs. 1.8%, *p* = 0.049), cancer (15.3% vs. 8.9%, *p* = 0.043), interstitial lung diseases (27.1% vs. 12.3%, *p* < 0.001), vasopressor use (11.8% vs. 4.5%, *p* = 0.002), emergency hemodialysis (2.4% vs. 0.4%, *p* = 0.010), albumin (4.7% vs. 1.0%, *p* = 0.002), heparin (14.1% vs. 5.3%, *p* = 0.001), insulin (25.9% vs. 12.3%, *p* < 0.001), rhTM (5.9% vs. 0.7%, *p* < 0.001), peramivir (71.8% vs. 59.0%, *p* = 0.018), carbapenem (20.0% vs. 8.9%, *p* < 0.001), cephalosporin (3rd generation) (44.7% vs. 31.7%, *p* = 0.012), anti-MRSA drugs (3.5% vs. 0.7%, *p* = 0.003), and steroid use (29.4% vs. 20.1%, *p* = 0.036) differed significantly between the groups before adjustment. After adjustment, the baseline characteristics were similar between the two groups, except for hospital volume (10.97 ± 1.3 vs. 14.55 ± 0.5, *p* = 0.010), laninamivir (0.47% vs. 2.55%, *p* < 0.001), and cephalosporins (2nd generation) (0.47% vs. 1.71%, *p* = 0.019).

Table 2 presents the primary and secondary endpoints in the moderate group assessed using the IPTW method with estimated propensity scores. There were no significant differences between the azithromycin and non-macrolide groups regarding total mortality (16.56% vs. 11.45%, *p* = 0.516), 30-day mortality (14.89% vs. 9.23%, *p* = 0.477), 90-day mortality (15.06% vs. 11.16%, *p* = 0.622), duration of ICU management (0.050 ± 0.03 days vs. 0.069 ± 0.01 days, *p* = 0.553), and hospital stay (17.55 ± 2.9 days vs. 17.95 ± 0.4 days, *p* = 0.891). However, the mean IMV duration after day 8 was shorter in the azithromycin group than in the non-macrolide group (0.028 ± 0.02 days vs. 0.154 ± 0.05 days, *p* = 0.026). 

### 3.3. Severe Group

Of the 607 severe patients, 90 (14.8%) were treated with intravenous azithromycin at 500 mg/day for >1 d, and 517 (85.2%) were not treated with macrolides within 7 days of hospitalization. The C-statistic for the propensity score was 0.785. Table 3 presents the baseline characteristics of the patients before and after adjusting for confounders. Patients taking cephalosporins (2nd generation) were excluded from the study because there were no patients. The baseline variables including albumin (34.4% vs. 20.5%, *p* = 0.004), antithrombin III (12.2% vs. 5.0%, *p* = 0.008), heparin (62.2% vs. 46.6%, *p* = 0.006), immunoglobulin (14.4% vs. 6.2%, *p* = 0.006), insulin (57.8% vs. 37.7%, *p* < 0.001), red blood cell transfusion (22.2% vs. 11.6%, *p* = 0.006), rhTM (20.0% vs. 7.9%, *p* < 0.001), peramivir (86.7% vs. 74.1%, *p* = 0.010), carbapenem (41.1% vs. 27.7%, *p* = 0.010), anti-MRSA drug use (15.6% vs. 5.4%, *p* < 0.001), steroid use (62.2% vs. 39.5%, *p* < 0.001), number of IMV days (5.18 vs. 4.21, *p* < 0.001), and ICU admission rate (38.9% vs. 28.4%, *p* = 0.046) differed significantly between the groups before adjustment. However, after adjusting for confounders, the baseline patient characteristics were similar between the azithromycin and non-macrolide groups, except for maintenance hemodialysis (0.80% vs. 2.78%, *p* = 0.006).

Table 4 present the results of the primary and secondary endpoints in the severe group assessed using the IPTW method with estimated propensity scores. Patients treated with azithromycin had significantly lower 30-day mortality rates than those not treated with macrolides (26.49% vs. 36.65%, *p* = 0.038). There were no significant differences between the azithromycin and non-macrolide groups regarding total mortality (33.78% vs. 40.97%, *p* = 0.194), 90-day mortality (33.78% vs. 40.37%, *p* = 0.234), duration of ICU management (2.96 ± 0.4 days vs. 2.59 ± 0.2 days, *p* = 0.407), IMV (10.82 ± 1.2 days vs. 11.69 ± 0.8 days, *p* = 0.534), and hospital stay (28.32 ± 2.3 days vs. 27.33 ± 1.3 days, *p* = 0.699).

## 4. Discussion

In this study, we investigated the effects of administering intravenous azithromycin within 7 days of hospitalization using data from 5066 patients with influenza virus pneumonia registered in the nationwide administrative database of Japan. Azithromycin was used in 192 (3.79%) of the 5066 patients based on real-world data. The use of azithromycin was proportional to the severity of respiratory failure (mild group, 1.0%; moderate group, 3.1%; and severe group, 14.8%). 

This study also demonstrated that intravenous azithromycin improved the duration of IMV in patients in the moderate group and 30-day mortality in patients in the severe group. There were no significant differences between the groups regarding total mortality and 90-day mortality in the severe group, but the azithromycin group had numerically lower results. It has been suggested that azithromycin may be effective in patients with more severe influenza pneumonia.

The efficacy of azithromycin for critical lung diseases such as CAP, ARDS, and acute exacerbation of interstitial pneumonia has been reported. In the United States, the most common risk factor for ARDS is severe sepsis with a suspected pulmonary source (46%) [25]. In a prospective cohort study of patients with severe CAP, the combination of beta-lactam and azithromycin was also associated with a significantly lower 30-day mortality (OR = 0.12, 95% confidence interval (CI): 0.007–0.57) compared with beta-lactam alone [26]. A meta-analysis of beta-lactam and macrolide combination therapy for CAP discovered that azithromycin-based combinations had higher clinical success rates than clarithromycin-based combinations (87.55% vs. 75.42%) [27]. A multicenter, multinational observational study of patients with severe CAP suggested that the combination of aspirin with CAM or azithromycin was associated with the 30-day survival rate (HR, 0.71, 95% CI: 0.58–0.88, *p* = 0.002) [28]. Influenza A viruses have also been reported to be a common cause of viral pneumonia and ARDS in adult patients, with ARDS observed in 39.4% of severe influenza cases [7]. In a report that assessed only patients with ARDS associated with viral pneumonia, 17% had ARDS associated with the influenza virus [29]. A secondary analysis of a multicenter RCT of patients with ARDS showed that macrolide use, including azithromycin, was associated with a lower 180-day mortality (HR, 0.46, 95% CI: 0.23–0.92, *p* = 0.028) and shorter time to successful weaning from mechanical ventilation (HR, 1.93, 95% CI: 1.18–3.17, *p* = 0.009) [16]. In a retrospective cohort evaluation of azithromycin’s effect on moderate or severe ARDS, azithromycin use was associated with a significant improvement in the 90-day survival rate (HR, 0.49, 95% CI: 0.27–0.87, *p* = 0.015) and shorter time to successful weaning from mechanical ventilation (HR, 1.74, 95% CI: 1.07–2.81, *p* = 0.026) [30]. Moreover, azithromycin significantly improved mortality in two studies on interstitial pneumonia. In a prospective, open-label study of acute exacerbations of chronic fibrosing interstitial pneumonia, the mortality rate was significantly lower in the azithromycin group than in the fluoroquinolone group (HR, 0.17, 95% CI: 0.05–0.61) [31]. In a retrospective study of acute exacerbations of idiopathic pulmonary fibrosis, the mortality rate was significantly lower in patients treated with azithromycin than in those treated with fluoroquinolones (26% vs. 70%, *p* < 0.001) [32]. These results suggest that the favorable effects of azithromycin on critical respiratory diseases may be due to its antimicrobial and anti-inflammatory effects.

The efficacy of macrolide antibiotics in patients with influenza has been reported in a few RCTs. Clarithromycin, a macrolide, in combination with naproxen and oseltamivir, has been associated with reduced 30-day mortality (*p* = 0.01), ICU admission rate (*p* = 0.009), and hospital stay (*p* < 0.0001) compared with oseltamivir alone [33]. The combination of clarithromycin and oseltamivir has also been shown to be more effective in improving clinical symptoms, such as rhinorrhea and fever, compared to oseltamivir alone [34,35]. In an RCT of patients with influenza, combination therapy with oseltamivir and azithromycin showed significant improvement in the levels of inflammatory cytokines, such as IL-6, CXCL8/IL-8, IL-17, CXCL9/MIG, sTNFR-1, and IL-18 [17]. Another RCT showed no improvement in inflammatory cytokines but faster improvement in clinical symptoms such as fever after day 4 in the oseltamivir and azithromycin groups (*p* = 0.048) [36]. Our results suggest that azithromycin may have favorable effects in patients with influenza and respiratory failure, consistent with these reports’ findings. However, another observational study showed no improvements in mortality with macrolides (clarithromycin, erythromycin, and azithromycin) in patients with influenza and severe respiratory failure [37]. Therefore, further studies, including RCTs, are warranted.

Macrolides have been demonstrated to possess some anti-inflammatory activity both in vitro and in vivo. Indeed, macrolides could modulate cytokine production as well as other immunological cellular properties (e.g., chemotaxis, degranulation, oxidative burst, and even apoptosis). Importantly, with regard to infectious diseases, the clinical relevance of macrolides may be beyond their antibacterial activity. They may improve the clinical course of viral respiratory infections (respiratory syncytial virus, influenza viruses, and coronavirus) at least through indirect mechanisms relying on variable anti-inflammatory activity [38]. Azithromycin, one of the macrolide antibiotics, is a low-cost, historically safe, and globally distributed antibiotic that is practical [39]. Azithromycin has immunomodulatory activity in the acute phase of inflammation suppression and late phase of chronic inflammation elimination [40]. The half-life is approximately 50 h, which is longer than that of other antibacterial agents, and such effective levels can be maintained for several days. Therefore, early administration can be expected to maintain antibacterial and anti-inflammatory effects during the acute phase. The effects of these immunomodulatory activities are due to the inhibition of extracellular signal-regulated kinase 1/2 phosphorylation and nuclear factor kappa B activation [40]. In vitro studies have demonstrated the ability of azithromycin to reduce the production of proinflammatory cytokines, such as IL-8, IL-6, and tumor necrosis factor-α; reduce oxidative stress; and regulate T-helper functions [41]. 

Azithromycin has also been shown to inhibit influenza and COVID-19 activity without cytotoxicity. It may be a candidate broad-spectrum antiviral drug [42]. Azithromycin has a significant anti-inflammatory effect on influenza infections [17], and there are reports showing antiviral effects that suppress the in vitro growth of many viruses [43]. However, the efficacy of azithromycin against COVID-19 has been negated in several RCTs such as the RECOVERY and ATOMIC2 trials. In the RECOVERY trial, which compared usual care alone to usual care plus azithromycin (500 mg oral/intravenous once daily), there were no significant differences in the 28-day all-cause mortality (rate ratio = 0.97, 95% CI: 0.87–1.07, *p* = 0.50), duration of hospital stay (median [interquartile range], 10 days [5 to >28 days] and 11 days [5 to >28 days], respectively), or proportion of patients discharged from the hospital alive within 28 days (rate ratio = 1.04, 95% CI: 0.98–1.10, *p* = 0.19) [44]. In the ATOMIC2 trial, which compared azithromycin (500 mg oral once daily for 14 days) plus standard care to standard care alone in 292 outpatients with mild to moderate illness, the 28-day mortality or hospitalization rates were 10% in the azithromycin group and 12% in the standard care group (adjusted OR = 0.91, 95% CI: 0.43–1.92, *p* = 0.80) [45]. However, many of these studies included patients with mild to moderate illness, and few studies evaluated only severe patients with respiratory failure or ARDS. As aforementioned, azithromycin has been reported to be effective against severe lung diseases such as CAP, ARDS, and acute exacerbation of interstitial pneumonia, and its effects in these populations with mild to moderate disease may have been underestimated. 

In the moderate group, azithromycin was used in combination with 3rd generation cephalosporin antibiotics (44.7%), penicillin antibiotics (37.6%), and carbapenem antibiotics (20.0%). In the severe group, 3rd generation cephalosporin antibiotics (34.4%), penicillin antibiotics (34.4%), and carbapenem antibiotics (41.1%) were used in combination, and carbapenem antibiotics were more frequently used in the severe group than in the moderate group. This included changes of antibiotics within 7 days and the use of three or more antibiotics at the same time, although azithromycin was considered to be administered in combination with other antibiotics in most patients. Similarly, antibiotics were used in most patients in the non-macrolide group. The reason for this is considered to be the effect of secondary bacterial infections associated with influenza. Approximately 23% of patients with influenza develop secondary bacterial infections associated with influenza, and secondary pneumonia complications are associated with increased mortality [46]. When the pandemic of influenza A occurred in 2009, bacterial infections occurred in approximately 30% of the patients who died of influenza [47]. According to a summary of clinical reports, 73.5% of all patients with viral pneumonias, including influenza, received prophylactic antibiotics [48]. Thus, the use of antibiotics in severe cases of pneumonia with influenza seems to be considered in the clinical setting. The Japanese guidelines for pneumonia also recommend macrolide antibiotics in combination with conventional antibiotics for severe pneumonia [49]. *Streptococcus pneumoniae*, *Haemophilus influenzae*, and *Staphylococcus aureus* are the most common causes [50]. Particularly, *Streptococcus pneumoniae* has a lethal synergy with the influenza virus [51]. In a retrospective cohort study investigating the effect of azithromycin on pneumococcal pneumonia, treatment with azithromycin was associated with a lower rate of mortality compared with other treatment groups that did not receive macrolide antibiotics (OR, 0.26, 95% CI: 0.08–0.80, *p* = 0.018) [52]. Data from the United States show that 34% of pneumococcal strains are penicillin insensitive (15.7% intermediate, 18.5% resistant) and the rate of macrolide resistance is 26.9–28.6%, indicating that antibiotic resistance is a problem [53]. On the other hand, azithromycin has been reported to be effective against macrolide-resistant pneumococcal pneumonia. In pneumococcal pneumonia requiring ICU management, 60.8% of patients had macrolide resistance, but azithromycin still reduced mortality (OR, 0.27, 95% CI: 0.09–0.85, *p* = 0.024) [54]. Azithromycin was effective in six of seven patients with high-level resistance (minimum inhibitory concentration > 256 μg/mL) [55].

Collectively, our favorable results against severe influenza might be due to the anti-inflammatory and antiviral effects of azithromycin and its antimicrobial effects in complicated bacterial infections. These effects can also be expected for macrolide antibiotics other than azithromycin, which were excluded from this study. However, further studies are needed to clarify these points [56].

Azithromycin is a very valuable antibiotic that is on the World Health Organization’s list of essential medicines and is required for the treatment of trachoma, multidrug-resistant tuberculosis, non-tuberculous mycobacteria, bacterial pneumonia, sexually transmitted diseases, and other diseases. According to Japan’s National Action Plan on Antimicrobial Resistance (https://www.mhlw.go.jp/content/10900000/000885373.pdf (accessed on 7 May 2023)), the macrolide resistance rate of *Streptococcus pneumoniae* was about 80% in Japan. As the results of microbiological examinations are not included in the DPC database, we could not evaluate the effects of increases in drug-resistant bacteria due to the use of azithromycin. In this study using a Japanese nationwide administrative database, azithromycin was mainly used in critical cases. The use of azithromycin was proportional to the severity of respiratory failure (mild group, 1.0%; moderate group, 3.1%; and severe group, 14.8%), and the effects were shown in the 30-day mortality rate of the severe group and the duration of invasive mechanical ventilation after day 8 of the moderate group. Accordingly, it might be better if concomitant use of azithromycin were limited to critical cases, for avoiding increases in drug-resistant germs, though further studies are needed.

This study had limitations similar to those of previous studies that used the DPC database [13,20,21]. First, this was an observational and retrospective study without randomization. We used the IPTW method with the estimated propensity score to reduce the effect of this limitation, although there were still significant differences in confounders. Such confounders included maintenance hemodialysis in the severe group and hospital volume, laninamivir, and cephalosporins (2nd generation) in the moderate group. Regarding maintenance dialysis in the severe group, azithromycin may have been avoided for use in patients with maintenance dialysis because it requires more fluid infusion, usually 500 mL at a time. Second, we could not include clinical data, such as peripheral blood laboratory findings, physiological data including vital signs, radiological findings, and mechanical ventilation settings. Third, it was difficult to analyze data including the content of previous treatments, such as the rate of multiple hospitalizations and previous history of antibiotic use, because of the difficulty in linking DPC data. Fourth, the influence of complicated bacterial infections was unknown because the results of microbiological examinations were not included in the DPC database. Thus, it was difficult to speculate whether the favorable effects of azithromycin were due to anti-inflammatory or antimicrobial effects. However, many of the patients included in this study were administered antibiotics other than azithromycin, and it can be inferred that there was no difference in the antibacterial effect between the two groups. Therefore, they may be due to the anti-inflammatory effects of azithromycin. Fifth, we were unable to evaluate adverse events of azithromycin. Azithromycin and other macrolide antibiotics cause serious adverse events, such as fatal arrhythmia and myocardial infarction, but this study could not evaluate these effects [57]. Additionally, we could not know the effects of increases in drug-resistant bacteria, as described above.

Despite these limitations, this study has the advantage of using real-world data and including a large number of patients (more than 5000) with influenza virus pneumonia and acute respiratory failure. In addition, data from April 2016 to March 2018 were not affected by the COVID-19 pandemic.

## 5. Conclusions

Using nationwide administrative database data, we discovered the favorable effects of intravenous azithromycin administration in patients with influenza virus pneumonia on mechanical ventilation or oxygen, based on propensity score analysis. The significance of this study is that it uses real-world data from a large number of patients with influenza virus pneumonia and acute respiratory failure, which is difficult to achieve in large prospective studies. Nevertheless, this study had some limitations owing to its retrospective and observational design, and well-designed RCTs are required to evaluate the efficacy of azithromycin further.

## Figures and Tables

**Figure 1 viruses-15-01142-f001:**
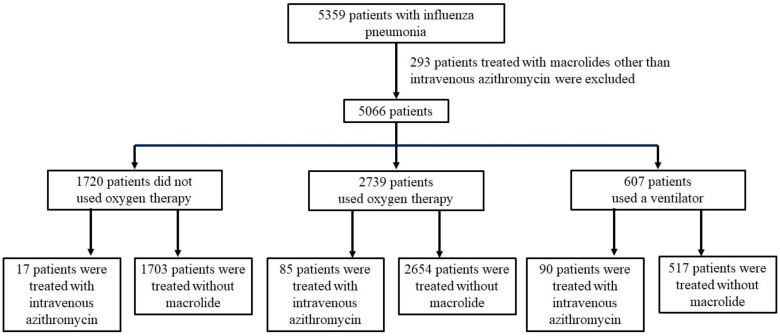
Flow chart of patient selection. Patients were classified according to treatment or no treatment within 7 days of hospitalization.

**Table 1 viruses-15-01142-t001:** Baseline characteristics of the patients in the moderate group with or without intravenous azithromycin administration before and after adjusting for confounders using the propensity score weighting method.

	Before Adjustment			After Adjustment		
Variables	Intravenous Azithromycin (n = 85)	Without Macrolide (n = 2654)	*p*-Value	Intravenous Azithromycin (n = 85)	Without Macrolide (n = 2654)	*p*-Value
Sex (female)	37.6	45.5	0.151	33.39	45.29	0.061
Age (years)	72.38 ± 2.1	70.84 ± 0.5	0.087	67.92 ± 3.2	70.87 ± 0.5	0.353
Hospital volume per year	11.79 ± 0.9	14.65 ± 0.5	0.092	10.97 ± 1.3	14.55 ± 0.5	0.010
Emergency admission	78.8	68.2	0.037	73.76	68.54	0.469
Emergency transport	65.9	55.3	0.054	60.85	55.60	0.488
Home healthcare	4.7	10.2	0.098	6.17	10.00	0.206
Smoking	50.6	32.7	0.001	29.17	33.20	0.464
Diabetes mellitus	18.8	17.0	0.652	13.59	17.00	0.441
Chronic kidney disease	12.9	8.2	0.121	5.91	8.34	0.350
Liver dysfunction	4.7	1.8	0.049	4.78	1.91	0.353
Cerebrovascular disease	5.9	10.7	0.155	18.53	10.55	0.350
Heart failure	12.9	15.4	0.534	10.72	15.33	0.218
Cardiovascular disease	4.7	4.2	0.827	3.89	4.23	0.911
Hypertension	24.7	25.4	0.892	20.26	25.37	0.307
Cancer	15.3	8.9	0.043	5.92	9.10	0.161
COPD	5.9	7.4	0.593	6.97	7.35	0.914
Asthma	12.9	13.7	0.846	11.52	13.65	0.495
Interstitial lung disease	27.1	12.3	<0.001	16.28	12.77	0.424
Chronic respiratory failure	4.7	1.8	0.060	1.72	1.92	0.770
Neurological dysfunction	2.4	2.7	0.856	9.78	2.68	0.361
Vasopressor	11.8	4.5	0.002	13.82	4.84	0.267
Maintenance hemodialysis	1.2	2.3	0.494	0.72	2.26	0.319
Emergency hemodialysis	2.4	0.4	0.010	0.31	0.48	0.270
Platelet transfusion	0.0	0.0	-	0.0	0.0	-
Red blood cell transfusion	2.4	1.6	0.601	2.96	1.68	0.682
Antithrombin III	0.0	0.0	-	0.0	0.0	-
rhTM	5.9	0.7	<0.001	2.15	0.94	0.391
Heparin	14.1	5.3	0.001	8.27	5.66	0.537
Insulin	25.9	12.3	<0.001	12.98	12.74	0.951
Albumin	4.7	1.0	0.002	1.00	1.17	0.751
Sivelestat	0.0	0.0	-	0.0	0.0	-
Immunoglobulin	0.0	0.0	-	0.0	0.0	-
Oseltamivir	9.4	10.4	0.769	8.25	10.35	0.579
Laninamivir	1.2	2.6	0.413	0.47	2.55	<0.001
Peramivir	71.8	59.0	0.018	63.22	59.36	0.635
Penicillin antibiotics	37.6	37.1	0.920	39.16	37.15	0.808
Cephalosporin (1st generation, iv)	1.2	1.0	0.857	0.72	0.99	0.858
Cephalosporin (2nd generation, iv)	1.2	1.7	0.697	0.47	1.71	0.019
Cephalosporin (3rd generation, iv)	44.7	31.7	0.012	27.29	32.06	0.415
Cephalosporin (4th generation, iv)	3.5	1.4	0.096	1.04	1.40	0.573
Carbapenem	20.0	8.9	<0.001	11.67	9.31	0.639
Clindamycin	1.2	0.5	0.339	0.51	0.48	0.919
New quinolone	5.9	5.7	0.928	14.90	5.68	0.121
Anti-MRSA drug	3.5	0.7	0.003	1.51	0.77	0.712
Minocycline	4.7	1.9	0.065	2.80	1.97	0.437
Steroid use	29.4	20.1	0.036	16.25	20.40	0.376
PMX	0.0	0.0	-	0.0	0.0	-
ICU admission rate	2.4	1.0	0.236	0.76	1.06	0.583

Data are presented as the % or the mean ± standard error, unless otherwise stated. The groups are adjusted using the inverse probability of treatment weighting method [24]. Abbreviations: COPD, chronic obstructive pulmonary disorder; IMV, invasive mechanical ventilation; NPPV, non-invasive positive pressure ventilation; oral, oral administration; PMX, polymyxin B-immobilized fiber column; rhTM, recombinant human soluble thrombomodulin; MRSA, methicillin-resistant *Staphylococcus aureus*; ICU, intensive care unit; iv, intravenously.

**Table 2 viruses-15-01142-t002:** Primary and secondary endpoint estimations using the propensity score weighting method in patients in the moderate group.

Outcomes	Intravenous Azithromycin (n = 85)	Without Macrolide (n = 2654)	*p*-Value
30-day mortality (%)	14.89	9.23	0.477
90-day mortality (%)	15.06	11.16	0.622
In-hospital mortality (%)	16.56	11.45	0.516
ICU management, mean days (±SE)	0.050 ± 0.03	0.069 ± 0.01	0.553
IMV, mean days (±SE)	0.028 ± 0.02	0.154 ± 0.05	0.026
Hospital stay, mean days (±SE)	17.55 ± 2.9	17.95 ± 0.4	0.891

Data are presented as the mean ± standard error. The groups are adjusted using the inverse probability of treatment weighting method. Abbreviations: IMV, invasive mechanical ventilation; ICU, intensive care unit; SE, standard error.

**Table 3 viruses-15-01142-t003:** Baseline characteristics of the patients in the severe group with intravenous administration of azithromycin or without macrolide before and after adjustment for confounders using the propensity score weighting method.

	Before Adjustment			After Adjustment		
Variables	Intravenous Azithromycin (n = 90)	Without Macrolide (n = 517)	*p*-Value	Intravenous Azithromycin (n = 90)	Without Macrolide (n = 517)	*p*-Value
Sex (female)	34.44	38.49	0.465	43.14	38.24	0.424
Age (years)	67.99 ± 1.7	68.03 ± 1.0	0.075	69.91 ± 1.7	68.05 ± 0.9	0.282
Hospital volume per year	9.19 ± 0.8	11.85 ± 0.8	0.433	10.11 ± 1.1	11.40 ± 0.7	0.298
Emergency admission	90.0	89.17	0.814	92.07	89.49	0.295
Emergency transport	78.9	75.0	0.434	80.81	75.92	0.101
Home healthcare	4.4	9.5	0.119	10.12	8.71	0.775
Smoking	51.1	47.8	0.559	50.21	48.83	0.727
Diabetes mellitus	24.4	19.5	0.285	23.32	20.35	0.539
Chronic kidney disease	17.8	15.7	0.614	13.23	15.87	0.440
Liver dysfunction	5.6	2.1	0.061	3.27	2.56	0.673
Cerebrovascular disease	8.9	8.7	0.954	11.02	8.76	0.556
Heart failure	15.6	23.6	0.091	26.54	22.50	0.541
Cardiovascular disease	6.7	6.4	0.919	51.49	63.06	0.519
Hypertension	18.9	21.9	0.526	27.33	21.29	0.368
Cancer	7.8	5.2	0.331	8.68	5.79	0.295
COPD	2.2	7.0	0.087	4.35	6.25	0.336
Asthma	4.4	6.6	0.441	3.90	6.17	0.130
Interstitial lung disease	20.0	15.9	0.329	12.90	16.19	0.241
Chronic respiratory failure	1.1	3.3	0.261	1.71	2.96	0.419
Neurological dysfunction	14.4	22.4	0.087	17.70	21.45	0.335
Vasopressor	63.3	56.3	0.212	66.75	57.89	0.158
Maintenance hemodialysis	1.1	3.1	0.293	0.80	2.78	0.006
Emergency hemodialysis	1.1	3.1	0.293	1.57	2.79	0.443
Platelet transfusion	11.1	7.4	0.222	8.26	7.82	0.835
Red blood cell transfusion	22.2	11.6	0.006	15.57	13.60	0.530
Antithrombin III	12.2	5.0	0.008	7.92	6.13	0.491
rhTM	20.0	7.9	<0.001	10.56	9.41	0.667
Heparin	62.2	46.6	0.006	53.53	48.76	0.434
Insulin	57.8	37.7	<0.001	47.64	40.85	0.220
Albumin	34.4	20.5	0.004	26.11	22.92	0.579
Sivelestat	5.6	2.9	0.193	55.85	34.58	0.539
Immunoglobulin	14.4	6.2	0.006	9.85	7.45	0.428
Oseltamivir	6.7	6.2	0.863	4.68	6.12	0.283
Laninamivir	1.1	1.2	0.968	0.47	1.10	0.078
Peramivir	86.7	74.1	0.010	82.26	76.05	0.153
Penicillin antibiotics	34.4	34.4	0.998	37.27	34.46	0.589
Cephalosporins (1st generation, iv)	2.2	2.1	0.954	2.15	2.03	0.924
Cephalosporins (2nd generation, iv)	0.0	0.0	-	0.0	0.0	-
Cephalosporins (3rd generation, iv)	34.4	33.3	0.827	33.35	33.32	0.995
Cephalosporins (4th generation, iv)	3.3	2.3	0.023	2.22	2.42	0.830
Carbapenem	41.1	27.7	0.010	35.96	30.12	0.337
Clindamycin	1.1	1.2	0.968	0.94	1.12	0.723
New quinolone	12.2	19.3	0.107	15.27	18.28	0.550
Anti-MRSA drug	15.6	5.4	<0.001	9.27	6.98	0.346
Minocycline	3.3	2.1	0.482	4.45	2.36	0.489
Steroid use	62.2	39.5	<0.001	45.43	42.78	0.599
PMX	1.1	0.6	0.566	0.45	0.62	0.471
Oxygen therapy	41.1	45.5	0.444	49.96	44.85	0.397
NPPV	5.6	4.3	0.581	3.77	4.25	0.750
IMV wearing days	5.18	4.21	<0.001	4.61	4.35	0.183
ICU admission rate	38.9	28.4	0.046	38.03	30.26	0.119

Data are presented as the % or the mean ± standard error, unless otherwise stated. The groups are adjusted using the inverse probability of treatment weighting method. Abbreviations: COPD, chronic obstructive pulmonary disorder; IMV, invasive mechanical ventilation; NPPV, non-invasive positive pressure ventilation; oral, oral administration; PMX, polymyxin B-immobilized fiber column; rhTM, recombinant human soluble thrombomodulin; MRSA, methicillin-resistant *Staphylococcus aureus*; ICU, intensive care unit.

**Table 4 viruses-15-01142-t004:** Primary and secondary endpoint estimations using the propensity score weighting method in patients in the severe group.

Outcomes	Intravenous Azithromycin (n = 90)	Without Macrolide (n = 517)	*p*-Value
30-day mortality (%)	26.49	36.65	0.038
90-day mortality (%)	33.78	40.37	0.234
In-hospital mortality (%)	33.78	40.97	0.194
ICU management, mean days (±SE)	2.96 ± 0.4	2.59 ± 0.2	0.407
IMV, mean days (±SE)	10.82 ± 1.2	11.69 ± 0.8	0.534
Hospital stay, mean days (±SE)	28.32 ± 2.3	27.33 ± 1.3	0.699

Data are presented as the mean ± standard error. The groups are adjusted using the inverse probability of treatment weighting method. Abbreviations: IMV, invasive mechanical ventilation; ICU, intensive care unit; SE, standard error.

## Data Availability

No new data were created or analyzed in this study. Data sharing is not applicable to this article.

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
