# Peer review of "Impact of Administering Intravenous Azithromycin within 7 Days of Hospitalization for Influenza Virus Pneumonia: A Propensity Score Analysis Using a Nationwide Administrative Database"

_viruses, 2023, doi:10.3390/v15051142_

Round 1

Reviewer 1 Report

Comments and Suggestions for Authors

The manuscript by Tokito et al. describes a retrospective observational study that uses the national Diagnostic Procedure Combination (DPC) in Japan to investigate potential health impacts of intraveneous azithromycin administration on hospitalized patients classified with mild, moderate, and severe influenza virus pneumonia. The authors used the inverse probability of treatment weighting (IPTW) to adjust for potentially confounding variables in their study population. Despite the limitations inherent in a retrospective study design, in contrast to a prospective randomized controlled trial, an advantage of this study was the very large study population of 5,066 patients. A significant difference in the primary endpoint of 30-day mortality rate was observed for the severe group, in which patients receiving IV azithromycin had a lower mortality rate than those patients who did not receive this treatment. Authors also observed a shorter mean duration of invasive mechanical ventilation for moderate group patients who received IV azithromycin. Overall, the findings of this retrospective observational study are generally consistent with previous reports and limited. Authors should clearly point out in the Discussion and Conclusions sections how the results of their study contribute to the advancement of the research in this area. Specific comments are provided below.

(1)   Materials and Methods:  In the patient selection subsection (starting at line 116), were all the patients included in the study population unique cases during the observational period or were some repeated/duplicated patients? If the latter, how were duplicated cases treated in this study?

(2)   Patients were selected in this study based on ICD-10 codes. How was influenza virus pneumonia verified in these cases? Were any of the cases complicated by secondary bacterial infections at the time of IV azithromycin treatment or was antibiotic treatment purely prophylactic?

(3)   Material and Methods (Variables subsection):  The authors considered “home healthcare” as a confounding variable (line 134). Was patient transfer to a hospital from a long-term care facility (e.g., nursing home) not considered a confounding variable, and if so, why not?

(4)   Table 1:  Please provide a reference for the IPTW method in the annotation for Table 1.  Also include potentially confounding variables that were adjusted for in the IPTW model.

(5) In the Conclusions section, authors should specifically and clearly state the findings of this study and their significance to the advancement of the science in this area.

Comments on the Quality of English Language

Overall, the quality of the writing meets publication standards with respect to accuracy and clarity.  Please correct the typo at line 222, so that it reads as follows:  "Table 2 presents the results...."

Reviewer 2 Report

Comments and Suggestions for Authors

The research presented in the manuscript is very important and useful. In general, I have no concerns about it. However, I have three questions and suggestions for refining the results of the work.

1. It is not specified how long (how many days) azithromycin was administered. The cumulative dose of the course of treatment may also be indicated.

2. It is not clear why the use of oral antibiotics was not assessed. Several antibiotics work as effectively in intravenous form when taken by mouth (oral form). If necessary, they are administered through a nasogastric tube.

3. It is not clear how it is possible that the number of "matched" patients remained the same as the number of "non-matched" patients.

It would be typical if 90 subjects in the „Intravenous azithromycin“ group (less than 90 would be likely to be matched) were matched with 90 (or fewer) subjects in the „Without macrolide“ group by all parameters (including hemodialysis).

It is extremely important to do this precisely. It is likely that with proper patient matching, the positive effect of azithromycin would no longer be found.

Reviewer 3 Report

Comments and Suggestions for Authors

Estimated Authors,

I've read with great interest your original article dealing with the effect of IV Azithromycin on the clinical outcome of patients hospitalized because of influenza virus infection.

Authors suggest that IV Azithromycin could reduce the 30-days mortality (26.5% vs. 36.7%), while other outcomes (e.g. in-hospital mortality) are way less affected by the treatment.

From my point of view, despite its potential interest, the paper would benefit from some significant adjustments before its acceptance.

1) Authors have preferred an approach based on propensity score weighting methods over matching of patients. PSWM has the considerable advantage over matching of including all the sampled patients, that are only (in brief) "weighted" up or down to make the patients in the treatment group and the comparison group more similar to each other. Authors should therefore explain, in methods section, why they opted for PSWM over matching. ON the contrary, limits of this approach are correctly reported in the discussion section, and no substantial adjustments are required.

2) Authors have reported all of the included variables before vs. after adjustment, but outcome variables are only reported as "after adjustment". In other words, please make consistent Table 4 with previous ones, including the estimates as before the adjustments.

3) Authors should discuss whether their results could be assumed as consistent with the post-pandemic landscape. In fact, during the early stages of the pandemic, an inappropriate use of antibiotics, including azithromycin, has been reported worldwide, and the AMR have exponentially increased. Therefore, some caveats on this specific topic should be included in the discussion section.

In the end (4) please adjust the very first section of the introduction in terms of phrasing. For example, "the influenza virus has become a seasonal epidemic ..." is quite inappropriate. More properly, "Seasonal influenza viruses causes seasonal epidemic, with around 290,000 to 640,000 respiratory deaths annually etc".

Comments on the Quality of English Language

The overall quality of the English text is acceptable, at least from my point of view. Only the phrasing of the first section should be amended in order to be more "plain"

Round 2

Reviewer 2 Report

Comments and Suggestions for Authors

None

Reviewer 3 Report

Comments and Suggestions for Authors

The present paper has been reworked according to my previous recommendations. Therefore, no further interventions are required.